# Re-evaluation of battery-grade lithium purity toward sustainable batteries

Gogwon Choe [1], Hyungsub Kim[2], Jaesub Kwon[1], Woochul Jung [3] ✉,
Kyu-Young Park [1,4] ✉ & Yong-Tae Kim [1,4] ✉

Recently, the cost of lithium-ion batteries has risen as the price of lithium raw materials has soared and fluctuated. Notably, the highest cost of lithium production comes from the impurity elimination process to satisfy the battery-grade purity of over 99.5%. Consequently, re-evaluating the impact of purity becomes imperative for affordable lithium-ion batteries. In this study, we unveil that a 1% Mg impurity in the lithium precursor proves beneficial for both the lithium production process and the electrochemical performance of resulting cathodes. This is attributed to the increased nucleation seeds and unexpected site-selective doping effects. Moreover, when extended to an industrial scale, low-grade lithium is found to reduce production costs and $CO_2$ emissions by up to 19.4% and 9.0%, respectively. This work offers valuable insights into the genuine sustainability of lithium-ion batteries.

Lithium-ion batteries (LIBs) have emerged as prevailing energy storage devices for portable electronics and electric vehicles (EVs) because of their exceptionally high-energy density compared with those of other energy storage systems[1]. However, the cost of LIBs, especially that of cathode materials, has been pointed out as the biggest hurdle to achieving affordable EVs. In this context, most cathode studies have been directed toward replacing expensive transition metals (TMs) with low-cost elements while increasing energy densities, for example, the introduction of disordered rocksalt[2,3], Co-free cathodes[4,5], high-Ni layered oxide[6–8], and over-lithiated layered oxide showing anionic redox[9,10]. These strategies have been praised for significantly reducing the cost of cathodes by removing expensive Co elements and enhancing their energy densities. Meanwhile, the lithium price has risen above eightfold from 2020 to the end of 2022 and fluctuated[11] (Fig. 1a) because lithium raw materials exist only in limited countries and due to the sudden changes in international situations[12]. Consequently, the lithium source cost now surpasses that of TM for new classes of cathode materials[13]. For example, as shown in Fig. 1b, the lithium material costs in $LiNi_{0.6}Co_{0.2}Mn_{0.2}O_2$ (NCM622) and $LiNi_{0.8}Co_{0.1}Mn_{0.1}O_2$

(NCM811) used to be 19.6% and 23.5% but are now up to 55.2% and 57.5%, respectively.

The soaring lithium costs naturally motivate us to take a closer look at lithium production, finding a chance to lower LIB prices. Most lithium sources are mined from lithium brines and hard rocks. As illustrated in Fig. 1c, these raw lithium materials are concentrated in liquid phases first. In the case of hard rocks, spodumene of $LiAl(SiO_3)_2$ requires calcination and acid roasting before the concentration process. Then, concentrated solutions are refined to meet the battery-grade purity of over 99.5% (yellowish region) followed by lithium extraction (LX) to produce final products such as $Li_2CO_3$ (LC) or $LiOH·H_2O$ (LH). Mg and Ca are the major impurities for the wet-chemical refinery process, and they are eliminated using the chemical agents $Ca(OH)_2$ and $Na_2CO_3$, respectively (the details of such processes are described in Supplementary Figs. S1 and S2 for brine and hard rock, respectively). It should be noted that these refinery processes are responsible for 30–40% of the total costs (Supplementary Tables S1 and S2) of lithium production as well as major $CO_2$ emission (see a yellowish region in Fig. 1c). Moreover, the removal of impurity precipitates in high-purity refining processes is costly due to the presence

[1]Department of Materials Science and Engineering, Pohang University of Science and Technology, 77 Cheongam-Ro, Nam-Gu, Pohang, Gyeongbuk 37673, Republic of Korea. [2]Neutron Science Division, Korea Atomic Energy Research Institute (KAERI), 111 Daedeok-daero 989 Beon-Gil, Yuseong-gu, Daejeon 34057, Republic of Korea. [3]Lithium Materials Research Group, Research Institute of Industrial Science and Technology (RIST), 67 Cheongam-Ro, Nam-Gu, Pohang, Gyeongbuk 37673, Republic of Korea. [4]Graduate Institute of Ferrous & Energy Materials Technology, Pohang University of Science and Technology, 77 Cheongam-Ro, Nam-Gu, Pohang, Gyeongbuk 37673, Republic of Korea. ✉e-mail: wcjung@rist.re.kr; kypark0922@postech.ac.kr; yongtae@postech.ac.kr

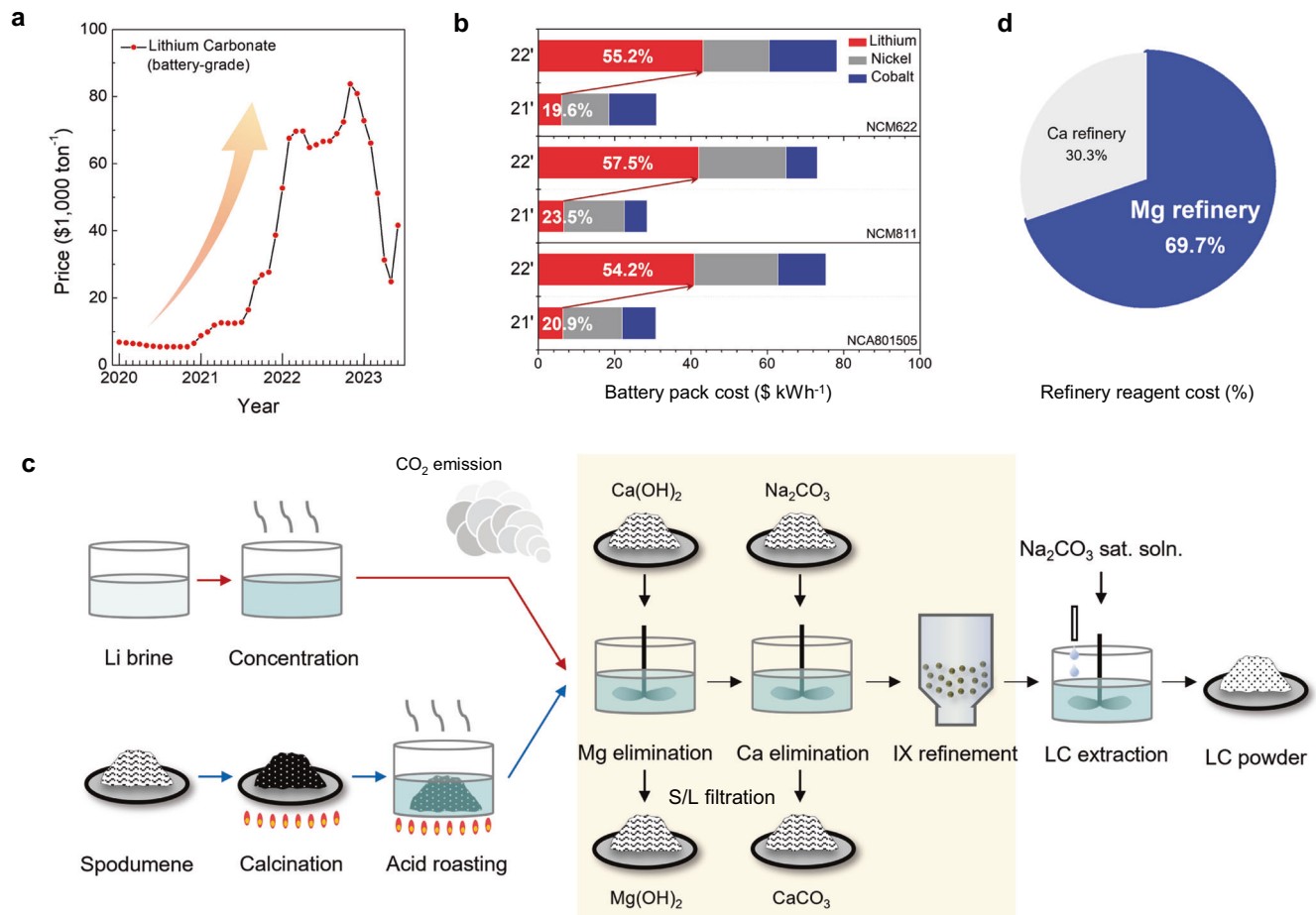

**Fig. 1 | Skyrocketing lithium prices and a scheme for lithium extraction processes. a** Price history of battery-grade lithium carbonate from 2020 to 2023[11]. **b** Cost breakdown of incumbent cathode materials (NCM622, NCM811, and NCA801505) for lithium, nickel, and cobalt based on material prices in March 2021 and 2022[13]. **c** Simplified process diagram of lithium carbonated production from lithium brine (top) and hard rock (bottom) (IX ion exchange, LC lithium carbonate, LX lithium extraction). **d** Reagent cost ratio of Mg and Ca refineries in lithium extraction processes.

of micron-sized particles. This necessitates expensive facilities, additional processing steps, and filter consumables. Notably, Mg elimination accounts for more than two-thirds of overall refinery costs compared with Ca refinery (Fig. 1d) because of Mg's smaller precipitate particles and high impurity concentrations. Accordingly, refinery works have dedicated themselves to achieving cost-effective and efficient Mg elimination[14–16] from high-Mg/Li-ratio brine[17,18] to meet the battery-grade purity. Furthermore, impurity and quality control in industry-scale cathode material production are gaining increased attention[19].

In this study, we systemically re-evaluated the impacts of impurity grade in terms of lithium extraction processes, electrochemical properties of the resulting cathodes, material production costs, and environmental impacts. We found that Mg impurity of up to 1% in lithium raw materials has unexpected benefits: (i) improvements in flowability and production speed of lithium product through the seeding effect, (ii) cyclability and rate capability enhancements through the anomalous site-selective doping effect, and (iii) significant reductions in expenditures and $CO_2$ emission ensuing from simpler purification processes. In particular, we would like to emphasize that the solid-solution phase of Li and Mg precursors increased the temperature at which the Mg dopant is incorporated into the oxide structure, leading to an abnormal Li site-selective doping. Consequently, notwithstanding the use of a low-grade lithium source, the electrochemical performances of the resulting cathode were superior to that of conventionally Mg-doped materials. Upon expansion to

industry-scale testing, lower-grade lithium sources reduced production costs and $CO_2$ emissions by 19.4% and 9.0%, respectively, because of the elimination of intensive precipitate removal processes and the decreased purity requirements. These new insights into the lithium grade should drive a paradigm shift toward securing the sustainability of LIBs.

## Results

### Production of *low-grade* lithium sources

We mimicked the conventional lithium extraction process from brine and hard rock but controlled the $Mg^{2+}$ impurity concentrations systematically to investigate their impact on lithium grade. Here, we produced a carbonate product rather than the hydroxide form as LH requires stringent storage conditions to control air exposure due to the carbonation reaction; in contrast, LC is more widely used as a strategic material in stages that require transport or storage. Bare LC, denoted as LCB, was synthesized by dropping a saturated $Na_2CO_3$ solution into a $Li_2SO_4$ solution batch, as shown in Supplementary Note S1. $Mg^{2+}$-containing LC powders were prepared in the same manner, but the reaction batches contained different amounts of $Mg^{2+}$ ions from 0.1 to 0.9 g $L^{-1}$ (Supplementary Note S1). Inductively coupled plasma optical emission spectrometry (ICP-OES; Supplementary Fig. S4 and Table S3) confirmed the grade of the prepared LC powders, showing a gradual Mg purity variation from 0% to 2.52% depending on the batch Mg concentration. We selected the 0.3 g $L^{-1}$ (0.98% Mg concentration in LC powder) and 0.9 g $L^{-1}$ (2.52% Mg concentration in

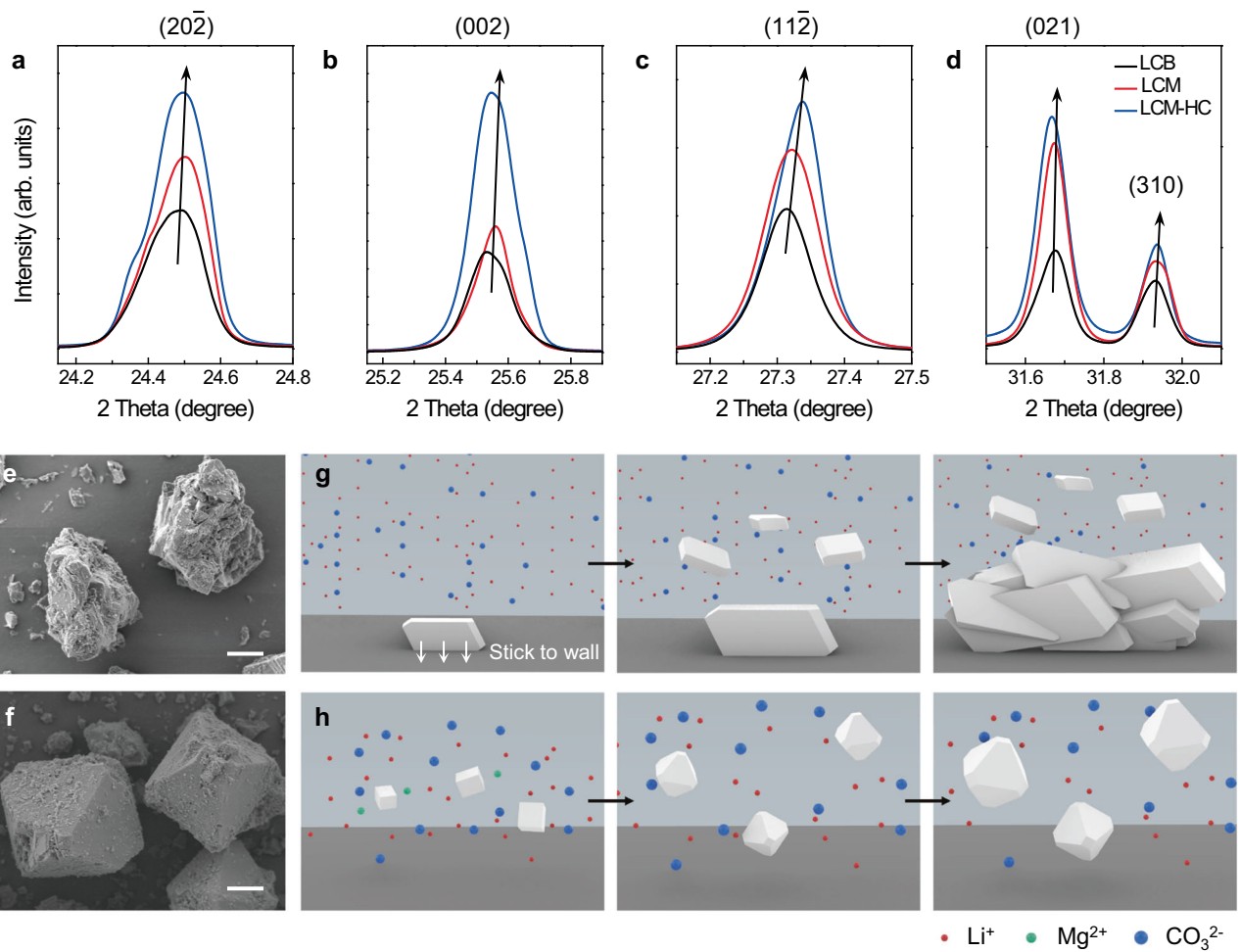

**Fig. 2 | Solid-solution behaviour and seeding effect of MgCO₃ for Li₂CO₃ synthesis. a–d** XRD patterns of the (20$\bar{2}$), (002), (11$\bar{2}$), (021) and (310) planes of LCB, LCM, and LCM-HC powders. High-angle shifts were observed at the (20$\bar{2}$) (**a**), (002) (**b**), (11$\bar{2}$) (**c**), and (310) (**d**) planes with the increasement of peak intensities.

**e, f** SEM images of LCB (**e**) and LCM-HC (**f**) particles. Scale bars, 10 μm. **g, h** Schematic diagrams of the nucleation and growth mechanisms of Li₂CO₃ without (**g**) and with (**h**) Mg²⁺ impurity.

LC powder) batch samples as representatives and denoted them as LCM and LCM-HC, respectively, for better discussion.

In this experiment, we found that the grade control of the lithium extraction process leads to the following distinguishable results: (i) formation of a MgCO₃–Li₂CO₃ solid solution, (ii) accelerated lithium extraction reaction rate, and (iii) change in the resulting particle morphology. A comparison of the X-ray diffraction (XRD) patterns of LCB, LCM, and LCM-HC showed a gradual high-angle shift of the Bragg peaks (Fig. 2a–d) with increasing Mg²⁺ concentration. They obeyed Vegard's law well, and a smaller ion radius of Mg²⁺ (0.72 Å) (Li⁺ = 0.76 Å) decreased the lattice parameters, suggesting the formation of the MgCO₃–Li₂CO₃ solid solution. However, a trace amount of crystalline secondary phase corresponding to MgCO₃·3H₂O was observed over the LCM condition, as discussed in Supplementary Note S2. In addition, a preferential increment of the peak intensity related to the *c*-axis was observed (Supplementary Fig. S6) according to impurity concentration, suggesting that the impurity promoted preferential crystal growth.

Mg impurity had a great advantage on the production rate of Li₂CO₃ through its seeding effect (Supplementary Fig. S7). While observing the precipitation process, we noticed that the mixed solution turned cloudy almost immediately after adding a Na₂CO₃-saturated solution to Mg²⁺-containing batches (Supplementary Fig. S7, black dashed line). In contrast, it took a few minutes for LCB to become cloudy (Supplementary Fig. S7, red dashed line). This supports that the

nucleation kinetics of Li₂CO₃ particles is better facilitated with Mg²⁺ ions because of the Mg²⁺ seeding effect as a result of the lower solubility of MgCO₃ than that of Li₂CO₃ (the solubilities of Li₂CO₃ and MgCO₃ were $6.9 \times 10^{-1}$ and $6.3 \times 10^{-3}$ g per 100 ml at 100 °C, respectively). The tracing of the pH change trend diagram well proves this seeding effect of the Mg²⁺ impurity. The diagram shows that the initial pH values were downshifted according to the concentration of Mg²⁺ impurity (Supplementary Fig. S7, red to purple line), indicating that the number of seeds (nucleation sites of Li₂CO₃) increased as the Mg²⁺ concentration increased, thus resulting in early-stage proton generation (see chemical reaction (5)).

The inclusion of impurity also led to well-defined particle morphologies (Fig. 2e, f) and narrowed particle size distributions (PSD) verified using scanning electron microscopy (SEM; Supplementary Fig. S8) and ex-situ PSD analysis (Supplementary Fig. S9). Generally, battery-grade Li₂CO₃ (Fig. 2g) slowly nucleated, particularly on the surface of the reaction batch and impeller, because of its lack of nucleation mediation. Primary particles tended to agglomerate and grow, making a secondary particle morphology with a needle-like primary particle shape (Fig. 2e) with inhomogeneous PSD[20]. Meanwhile, the introduction of Mg impurity resulted in an octahedron morphology (Fig. 2f and Supplementary Figs. S10 and S11), which supports the preferred crystal growth confirmed by the XRD patterns (Supplementary Fig. S6). For Li₂CO₃ synthesized with Mg²⁺ impurity (Fig. 2h), MgCO₃·*x*H₂O nanoparticles first precipitated because of its

lower solubility, providing a seeding effect to $Li_2CO_3$ nucleation. $Li_2CO_3$ nucleated and grew on the generated $MgCO_3 \cdot xH_2O$ seeds as an octahedron-like morphology, which is highly supposed to have resulted from the $MgCO_3 \cdot xH_2O$ seeds[21]. Morphology control is a critical factor in enhancing flowability and compressibility during wet-chemical powder production. As a result, numerous studies in lithium production have focused on achieving well-defined particle morphologies by reducing supersaturation[22–24] and adjusting crystal growth conditions[25]. Our observation suggests that Mg impurity is effective in controlling morphology as well, thus preventing lithium powders from clogging in the reaction batch and on the impeller (Supplementary Fig. S12). As such, Mg impurities provide an ample number of nucleation sites in the solution, which can help maintain enhanced flowability and compressibility, providing significant advantages to LX processes. Furthermore, the transition to a single-crystalline-like morphology increased the filling ratio of the furnace during the cathode material calcination process, thereby enhancing process capacity (Supplementary Fig. S13).

## Electrochemical performance verification

Conventionally, $Mg^{2+}$ doping on layered oxide cathode was implemented via two different ways: (i) the use of separated Mg precursor (denoted as solid-state doping)[26] or (ii) the use of the Mg-containing TM precursor prepared through coprecipitation (denoted as coprecipitation doping)[27]. Here, we compared the electrochemical performances of these two conventional $Mg^{2+}$ doping methods with a cathode from an impurity-adjusted lithium source (lithium-carbonate doping). All series of prepared Mg-containing $Li_2CO_3$ were used as lithium sources for synthesizing NCM622 cathodes, a widely used and representative layered-oxide-type cathode material. The powders were mixed with the transition metal precursor of $Ni_{0.6}Co_{0.2}Mn_{0.2}(OH)_2$ and calcinated (Supplementary Note S1; lithium-carbonate doping). The ICP-OES analysis (Supplementary Table S4) and XRD patterns (Supplementary Fig. S14) showed the exact $Mg^{2+}$ concentration well doped in these cathodes from 0.5 to 3.6 mol% without detectable impurity phases, and negligible differences in particle morphology were observed (Supplementary Fig. S15).

Preliminarily electrochemical tests exhibited improved performances in cycle retention from the doping effect[28] with up to 1.5 mol% Mg-doping concentration (from a 1 wt% Mg-impurity-containing LCM precursor, denoted as LCD hereinafter). However, the further inclusion of impurity rather led to decreased performances (Supplementary Fig. S16). Given that the 1.5 mol% Mg-doped NCM622 showed the best electrochemical performance, the comparative doping study depending on synthesis routes was implemented at a doping concentration of 1.5 mol%. The synthesis routes of solid-state doping and coprecipitation doping were denoted as SSD and CPD, respectively (see detailed material characterization in Supplementary Note S3).

The comparison of electrochemical measurements revealed an interesting variation in the performance of $Mg^{2+}$-doped NCM622 cathodes depending on the doping methods (Fig. 3). Figure 3a, b respectively, illustrate the galvanostatic charge/discharge profiles of the first and second cycles at a current density of 16 mA g$^{-1}$ in the voltage range of 3.0–4.3 V at 25 °C. The initial capacity was similar, with a slight decrease in the $Mg^{2+}$-doped cathodes, possibly attributable to the redox-inactive nature of $Mg^{2+}$. This consistency was observed at the higher temperature of 40 °C (Fig. 3c, d). All 1.5-mol% $Mg^{2+}$-doped cathodes exhibited improved cycling performances compared to the Bare cathode, with no significant initial capacity loss from multiple half-cell tests (Fig. 3e; error bars represent standard deviation). Notably, the LCD cathode achieved the highest capacity retention of 82.6%, while the SSD and CPD cathodes retained 80.9% and 78.8%, respectively, of their initial capacities after 200 cycles with a cutoff voltage of 4.3 V at a current density of 276.5 mA g$^{-1}$ at 20 °C. To further accentuate the differences in capacity retention performance among the

cathodes, high-temperature cycling tests were conducted at 40 °C accelerating cycling degradation. As shown in Fig. 3f, the LCD cathode retained over 60% of its capacity after 200 cycles, whereas the other cathodes reached 60% retention earlier. This cycling performance trend remained consistent with more pronounced differences at 40 °C. Additionally, the LCD cathode demonstrated superior rate capability compared with the other Mg-doping methods, as shown in rate capability tests conducted at various current densities with a cutoff voltage of 4.3 V at 25 °C (Fig. 3g). The full cells were constructed using commercial graphite as the anode to further verify the improved cycling performance of the LCD cathode (Fig. 3h). These full cells were cycled in the voltage range of 3.0–4.2 V (vs. graphite) at 40 °C, with a current density of 66.7 mA g$^{-1}$. The charge/discharge curves of the initial formation cycles of these full cells can be found in Supplementary Fig. 20. The comparable cathodes of LCD, SSD, CPD, and Bare retained 89.1%, 87.9%, 85.0%, and 80.8%, respectively, of their initial capacities after 200 cycles. The cathodes showed a consistent cycling performance trend in full cell composition, with LCD demonstrating the highest cycling stability. It should be highlighted that LCD, synthesized using a low-grade lithium source, outperformed other doping strategies without intensive impurity removal (further comparison studies for high-concentration Mg-doping are additionally conducted and described in Supplementary Fig. S19). Furthermore, the level of Mg impurity in the lithium precursor can easily be regulated by adjusting the refining process (see detailed explanation in Supplementary Note S4).

## Anomalous doping effect of low-grade lithium sources

The electrochemical comparison suggests subtle doping state differences in cathode materials depending on the doping methods. Accordingly, fine crystal structural analysis was conducted using neutron diffraction (ND) to find the origin of such differences. It must be noted that ND is suitable for doping site studies because of its capability of detecting lightweight elements, such as Li and Mg, through nuclear interaction scattering[29]. Supplementary Fig. S22 compares the measured whole ND patterns of the bare and 1.5-mol% $Mg^{2+}$-doped NCM622s. We further simulated ND pattern changes according to two possible crystallographic positions of $Mg^{2+}$, exclusively in Li layers (3a) and TM layers (3b), and compared them with measured patterns (Fig. 4a, b). The pattern simulations (Fig. 4a and Supplementary Fig. S23A) show the peak height changes of the low-index planes of (003), (101), and (012) depending on the $Mg^{2+}$ positions. With doping, the peak heights of the (003) and (101) planes became lower than those of the bare pattern regardless of positions, but the peak heights further weakened in the case of the 3a site doping. A notable difference was found in the (012) peak, where the height decreased (−4.18%) as $Mg^{2+}$ was located in the 3a site but increased (+3.11%) in the 3b site. Compared with the measured patterns, LCD followed more the tendency of the simulation result of $Mg^{2+}$ in the 3a site (Fig. 4b and Supplementary Fig. S23B), which showed the lowest peak heights at low-angle indices at (003), (101), and (012). Meanwhile, the SSD and CPD patterns matched more with the $Mg^{2+}$ in the TM site simulation results.

The further combination of XRD (Supplementary Fig. S24) and ND refinement analysis gave more solid evidence for the selective Mg-doping phenomenon. For increased accuracy, powder XRD refinement first provided the general information of the samples, such as the lattice parameters and atomic positions, and the following ND analysis strictly confirmed the atomic occupancy and thermal coefficient on the Li slab separating $Li^+$, $Mg^{2+}$, and $Ni^{2+}$ signals using their distinguishable atomic scattering in ND. Then, this XRD–ND iterative refinement was implemented until all refinement indicators were in reliable ranges. All Rietveld refinement results of the fitting data and detailed refinement parameters are in Supplementary Figs. S25 and S26 and Tables S8–S12. The most noticeable difference was that LCD

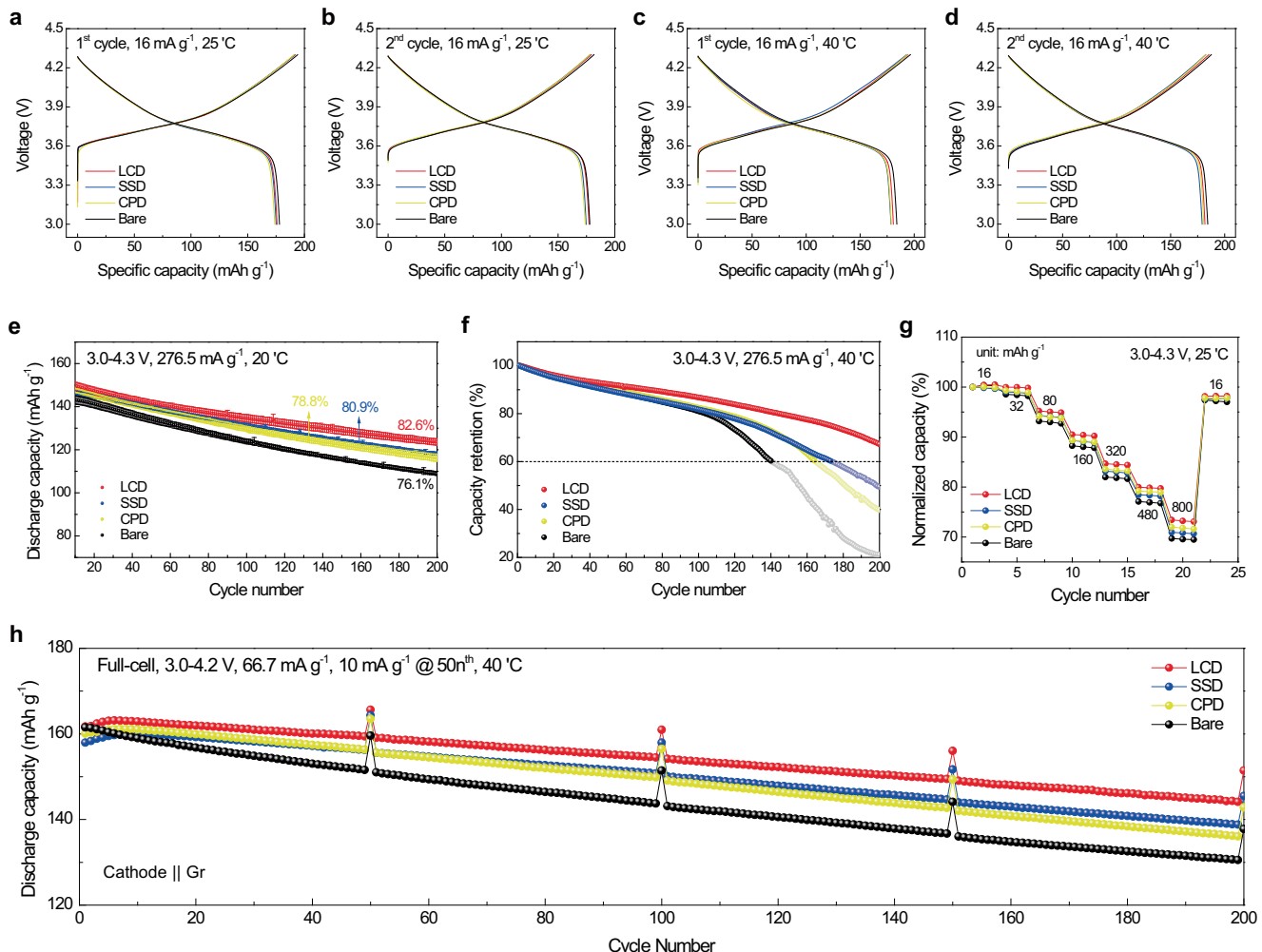

**Fig. 3 | Comparison of electrochemical performances depending on doping strategies. a–d** Charge/discharge curves of LCD, SSD, CPD, and Bare in the voltage range of 3.0–4.3 V at a current density of 16 mA g⁻¹; first cycle (**a**) and second cycle (**b**) at 25 °C, first cycle (**c**) and second cycle (**d**) at 40 °C. **e, f** Capacity fading of cathodes at a current density of 276.5 mA g⁻¹ with a cutoff voltage of 4.3 V at 20 °C (**e**) and 40 °C (**f**). **g** Rate capability of cathodes at various current rates of with a cutoff voltage of 4.3 V at 25 °C. **h** Capacity fading of cathodes at a current density of 66.7 mA g⁻¹ with a cutoff voltage of 4.2 V (vs. graphite) at 40 °C in the full cell. The capacities were checked every 50th cycle at a current density of 10 mA g⁻¹, and a constant voltage of 3.0 V was applied at the end of discharge (10 mA g⁻¹ current cutoff in the constant voltage).

showed the highest occupancy ratio of $Mg^{2+}$ in the Li site ($Mg_{Li}\%$), as summarized in Fig. 4c. More than 80% of doped $Mg^{2+}$ ions, corresponding to 1.2 mol%, occupied the Li site in LCD, whereas around 65% of $Mg^{2+}$ ions were found in the Li slab for both SSD and CPD. Furthermore, X-ray absorption results (Supplementary Fig. S27) supported the doping site selectivity of Mg depending on the synthetic route (more detailed discussions about other refinement parameters are in Supplementary Fig. S28). The $Mg^{2+}$ ions in Li layers resulted in the so-called pillaring effect between TM layers, which enhanced cyclability and rate capability (Fig. 3) by hindering the lattice collapse and increasing the *c*-axis spaces of Li diffusion layers, respectively[27,28,30,31]. Moreover, a previous study that selectively doped $Mg^{2+}$ ions into Li sites minimized the replacement of redox-active TMs with $Mg^{2+}$ and thus minimized the initial capacity loss[32]. These arguments will explain the superior electrochemical performances of LCD compared with those of other doping methods (additional structural studies of high-concentration $Mg^{2+}$-doped cathode materials can be found in Supplementary Note S5).

When the final doping state solely follows thermodynamics, the concentrations at certain doping sites would be identical, regardless of the synthesis methods. However, differences in precursor mixtures or material conditions can lead to variations in the final product results

due to distinct kinetic environments. For instance, recent research has reported that the kinetics of lithium transition metal layered oxide growth noticeably varied depending on the distribution of the Li source around the transition metal oxide host structure[33]. Such findings introduce a new perspective on how the thermal diffusion of doping or guest ions into the host structure can significantly influence the final phase. It is well known that the TM hydroxide precursor ($Ni_{0.6}Co_{0.2}Mn_{0.2}(OH)_2$) turns into a rocksalt structure of (Ni, Co, Mn)–O after the water inside the precursor evaporates. Thereafter, Li ions thermally diffuse into the rocksalt host structure above the melting point of the Li source (e.g., 723 °C for $Li_2CO_3$), which oxidizes TM and constructs an ordered layered structure of cathode material ($Li + Ni_{0.6}Co_{0.2}Mn_{0.2}O_2$)[34,35]. These previous insights naturally lead to the consideration of the thermal diffusion behaviour of doping and guest ions, with expected kinetic differences for $Mg^{2+}$ and $Li^+$ due to the melting points of $MgCO_3$ (350 °C) and $Li_2CO_3$ (723 °C), respectively. The thermal diffusion behaviour of SSD and LCD was confirmed through chemical composition analysis. The precursor mixture of SSD and LCD was calcined at 400 °C for 5 h, which was above the melting point of $MgCO_3$ but below the melting point of $Li_2CO_3$, in order to analyse the intermediate phase. The backscattered electron image (Supplementary Fig. S32) shows sufficient mechanical mixing of Li and

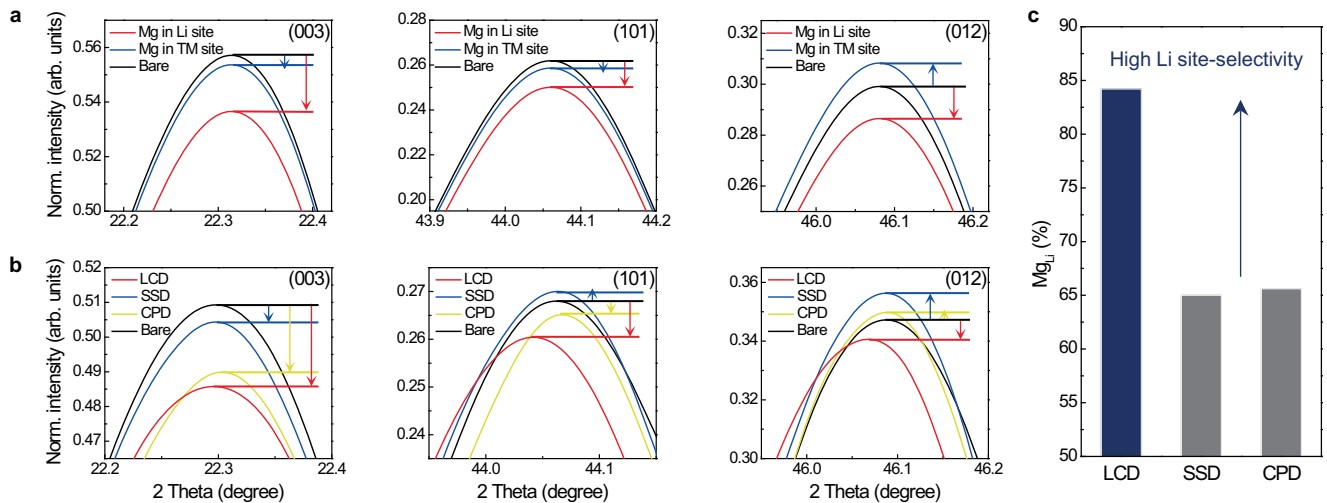

**Fig. 4 | Fine structural analysis defining the crystallographic position of Mg. a, b** Comparisons of simulated (**a**) and observed (**b**) ND patterns. **c** The ratio of Mg in the Li slab against the total amounts of Mg in the cathode materials was analysed through ND Rietveld refinement.

TM precursors. SEM-EDS analysis further confirmed that Mg was homogeneously distributed in all SSD particles (Supplementary Figs. S33 and S34), while Mg and TMs were distributed in different spatial regions in the LCD mixture under identical conditions (Supplementary Figs. S33 and S35). Additionally, the altered XRD pattern of the intermediated rocksalt phase provided clear evidence of the presence of Mg in the rocksalt phase of SSD (Supplementary Fig. S36). This observation directly demonstrates that the synthesis results could vary depending on which precursor contains the dopant.

On the basis of the above findings, we propose doping models depending on the sources of Mg (Fig. 5). Let us consider the doping processes of SSD and CPD incorporating $Mg^{2+}$ ions from separated or transition metal precursors, respectively. The formation of the (Ni, Co, Mn)−O rocksalt structure initially occurred at a certain temperature in the SSD mixture (Fig. 5a). Then, $MgCO_3$ is liquified above its melting point of 350 °C, and $Mg^{2+}$ ions diffused into (Ni, Co, Mn)−O to form the (Ni, Co, Mn, Mg)−O rocksalt phase. Likewise, the TM hydroxide precursor of CPD (Fig. 5b) formed the (Ni, Co, Mn, Mg)−O rocksalt structure as the water evaporated. The (Ni, Co, Mn, Mg)−O rocksalt phases in SSD and CPD constructed the final ordered layered phases above the melting point of the Li source as Li ions were thermally supplied and $Mg^{2+}$ ions were redistributed under the constructed layered structure.

Meanwhile, the LCD (Fig. 5c) case was unable to uptake $Mg^{2+}$ ions into its rocksalt (Ni, Co, Mn)−O phase until the Mg/Li source was liquified. The TM hydroxide precursor formed (Ni, Co, Mn)−O at an elevated temperature, but the solid solution of dilute $MgCO_3$ in $Li_2CO_3$ (Fig. 2a–d) more followed the melting behaviour of the major phase when considering the typical phase diagram of binary solid-solution systems (Supplementary Fig. S37). As the temperature increased near ~723 °C, $Li^+$ and $Mg^{2+}$ ions were then able to co-diffuse into the (Ni, Co, Mn)−O rocksalt structure simultaneously, forming the final layered phase where most of the $Mg^{2+}$ ions were positioned at the Li site.

According to a previous work, $Mg^{2+}$ ions thermodynamically prefer Li sites at concentrations below 2 mol%[30]. Nevertheless, the actual doping result varied sensitively depending on synthetic conditions. Thus, we must consider the melting point of the dopant precursor and the degree of distribution of the doping element. The site selectivity of the dopant was kinetically controlled by adjusting the incorporation timing of the doping element. Because of the solid-solution property of $Mg^{2+}$-containing $Li_2CO_3$, $Mg^{2+}$ could be trapped in the $Li_2CO_3$ until the layered phase evolution begins, whereas solid-

state doping and coprecipitation doping processed first the formatting of $Mg^{2+}$-containing rocksalt structure.

## Economic and environmental impacts

The inclusion of Mg impurity during lithium extraction improved the lithium extraction rate, powder flowability, and compressibility through controlled particle morphology. Moreover, the fine structural analysis demonstrated that Mg impurity at a level of about 1% in the lithium source can boost the electrochemical performance by maximizing the pillaring effect and increasing the layer space. This comprehensive re-evaluation of impurity grade provides us a chance to address the cost and environmental issues of the production of battery materials. In this regard, we calculated an annual plant production capacity of 25,000 tons of $Li_2CO_3$ per year. On this basis, economic and environmental evaluations were conducted for the introduction of 1% Mg impurity low-grade $Li_2CO_3$ (Table 1; more detailed calculation tables are in Supplementary Tables S13−S21). It is worth noting that the pilot plant (P/P) test conducted here utilized actual brine pumped from 'Hombre Muerto' Salt Lake in the Argentine highlands and spodumene mined from Western Australia of which results served as the basis for the evaluations.

The economic evaluation was performed in two respects: (i) capital expenditure (CAPEX); productivity and the scale of facilities and (ii) operating expenditure (OPEX). The CAPEX was mainly attributed to solid−liquid filtration (S/L filtration) equipment cost for eliminating $Mg(OH)_2$ by-products (Supplementary Tables S15 and S16). As shown in our P/P test (Supplementary Table S17), the Mg refinery generated 15.84 ton (LC ton)$^{-1}$ of $Mg(OH)_2$ cake, whereas the Ca refinery produced only 0.98 ton (LC ton)$^{-1}$ of $CaCO_3$. Moreover, as the D10 of the $Mg(OH)_2$ cake was generally obtained around 3 μm or below (Supplementary Fig. S38), this small-sized $Mg(OH)_2$ precipitation resulted in a cost issue in the S/L filtration process. The small amount of impurity was hard to remove using normal filter press equipment; thus, additional ion exchange filters (IX) or advanced disc filter processes are required as the next purifying step to achieve battery-grade purity. Here, however, the advanced filtration step is now able to be skipped with improved Mg impurity tolerance (~1%), which led to a considerable reduction in equipment cost for brine and hard rock processes of 12 and 9.5 million USD, corresponding to 19.4% and 7.3% reductions from total equipment cost, respectively (Table 1, Supplementary Tables S15 and S16).

With higher Mg impurity tolerance, the total OPEX costs were significantly reduced as well in both brine and hard rock processes by

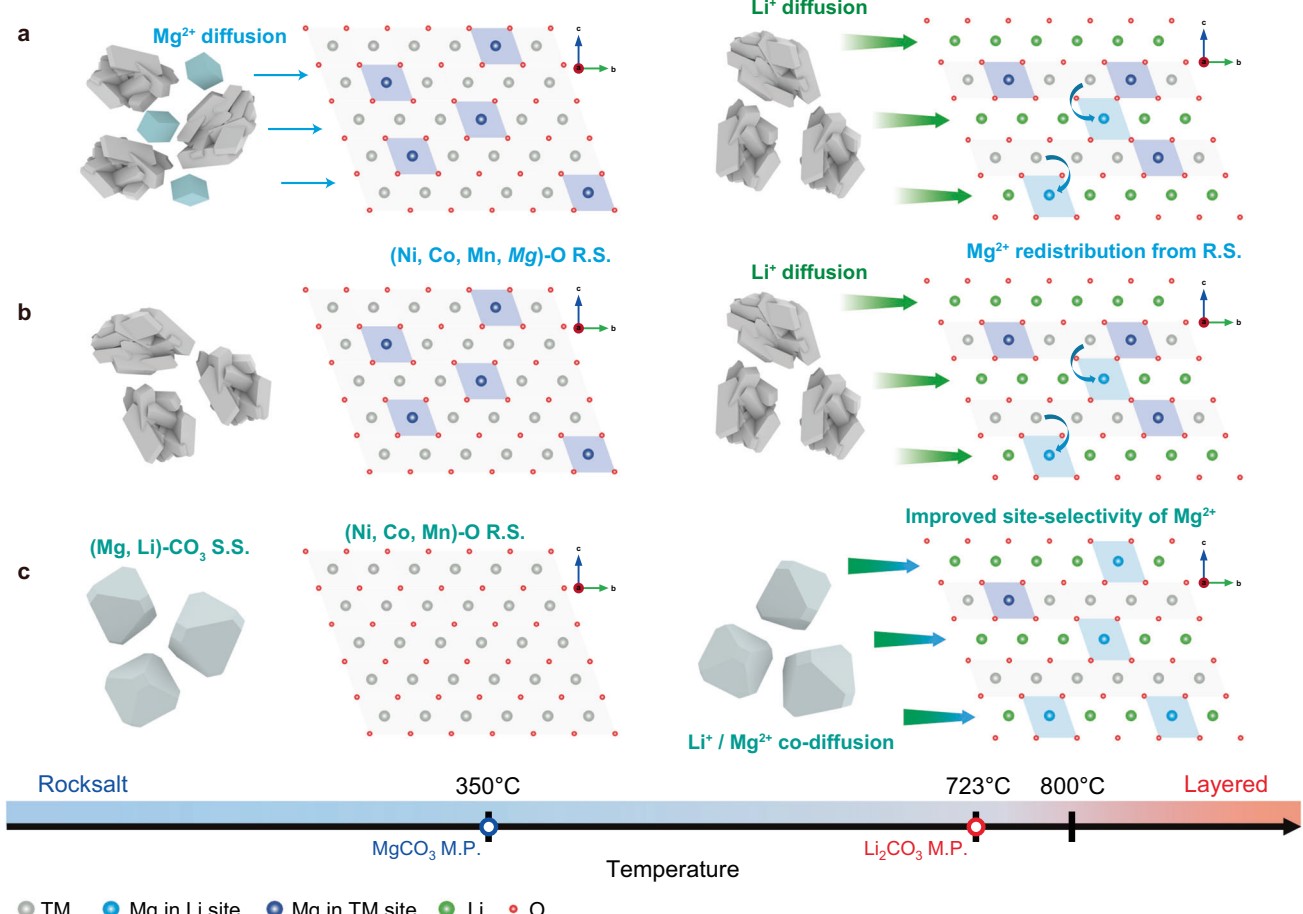

**Fig. 5 | Kinetically controlled site selectivity of Mg²⁺ by adjusting the incorporation timing. a–c** Schematic diagrams of the thermal diffusion pathways of Li⁺ and Mg²⁺ during cathode materials synthesis are shown. The TM hydroxide precursor turned to a rocksalt structure as the temperature increased. SSD (**a**) and CPD (**b**) first constructed a (Ni, Co, Mn, Mg)−O rocksalt phase, and Mg²⁺ were redistributed inside the layered structure as Li⁺ started to diffuse. **c** However, in the case of LCD, Li⁺ and Mg²⁺ co-diffused to the (Ni, Co, Mn)−O rocksalt structure, which improved the site selectivity of Mg²⁺ to the Li site.

188.3 and 246.1 USD (LC ton)⁻¹, respectively, which respectively correspond to 5.9% and 3.5% reduction from the total OPEX (Table 1). All consumables, reagents, utilities, and waste disposal costs were partially reduced with the decreased process load (see the details in Supplementary Tables S18 and S19). The major cut in cost was attributed to the less use of Mg elimination agents, which not only reduced reagent cost, but also reduced by-product leading to utility and waste disposal cost reduction. Moreover, from the P/P scale Mg refinery test

using brine from Argentina (Supplementary Tables S20 and S21), 79.9 USD of reagent cost was diminished per 1 ton of $Li_2CO_3$ production.

The environmental impact was evaluated with respect to $CO_2$ emission. As environmental, social, and governance pressures are ever-increasing worldwide, the reduction of the carbon footprint of the lithium production sector is highly critical for sustainable LIBs. Supplementary Tables S22 and S23 compare the carbon footprints of the conventional and less-purifying Mg LX processes for brine and hard rock, respectively. The Mg refinery reagent $Ca(OH)_2$ has a high carbon footprint because quicklime (CaO), the precursor of $Ca(OH)_2$, requires the high-temperature calcination of limestone and produces $CO_2$ as a product. Thus, the reduction of the usage of this reagent could curtail $CO_2$ emissions. Furthermore, reductions in the energy and waste carbon footprints were achieved by the IX process and $Mg(OH)_2$ cake reduction (see Supplementary Tables S22 and S23). Overall, the $CO_2$ emissions in the brine and hard rock LX processes were reduced by 9.0% and 4.4% of the total $CO_2$ emissions, respectively (Table 1), achieving a more sustainable LX process. However, the potential benefits from the enhancements in battery performance and filtration efficiency by $Li_2CO_3$ particle morphological effects were not considered in this analysis. Thus, additional reductions in production costs and carbon footprints are expected.

## Table 1 | Economic and environmental evaluation of overall processes

| LX process | | CAPEXᵃ (M $) [%] | OPEXᵇ ($/LC ton) [%] | CO₂ emissionᶜ (eq. ton/LC ton) [%] |
|---|---|---|---|---|
| Brineᵈ | As is | 61.8 | 3221 | 9.47 |
| | Reduced | 49.8 [−19.4%] | 3032.7 [−5.9%] | 8.615 [−9.0%] |
| Hard rockᵉ | As is | 133.6 | 6995 | 7.45 |
| | Reduced | 124.1 [−7.3%] | 6748.9 [−3.5%] | 7.125 [−4.4%] |

ᵃReduction in equipment cost by millions of USD and reduction ratio from the total equipment costs (Supplementary Tables S15 and S16).
ᵇReduction in OPEX in USD per 1 ton of LC production and reduction ratio from the total OPEX (Supplementary Tables S18 and S19).
ᶜReduction in $CO_2$ emission by equivalent ton per 1 ton of LC production and reduction ratio from the total $CO_2$ emission (Supplementary Tables S22 and S23).
ᵈBrine LX process (Supplementary Fig. S1).
ᵉHard rock LX process (Supplementary Fig. S2).

## Discussion

From salt lakes to batteries, the LIBs used today are undergoing intense scrutiny from the perspectives of sustainability and the pursuit

of higher energy density. In this work, we comprehensively re-evaluated the electrochemical, economic, and environmental impacts of the battery-grade purity of Li sources. We discovered that Mg impurities of about 1% have unique benefits in total production cost and $CO_2$ emission during LX processes. Moreover, the inclusion of $Mg^{2+}$ ions in the Li source resulted in the thermal diffusion of the dopant at a higher temperature, which reinforced the pillar effect and resulted in electrochemical advantages. Whereas many studies aimed to reduce the costs of TMs by controlling redox chemistry, we addressed the general belief on the battery-grade purity of Li sources and concluded that precursors with slight amounts of impurities can maximize the productivity and eco-friendliness of batteries. Hence, we suggest a reconsideration of the meaning behind industrial and commercial specifications on battery-grade purity to accomplish ultimate material stability for LIBs.

## Methods

### Materials synthesis

Lithium carbonate powder was synthesized via the precipitation method. Aqueous solutions with $10 g L^{-1}$ of $Li^+$ and different amounts of $Mg^{2+}$ (0.1, 0.3, 0.5, 0.7, and $0.9 g L^{-1}$) ions were prepared by dissolving $Li_2SO_4 \cdot H_2O$ and $MgSO_4$ with appropriate ratio, and sodium carbonate saturated solution was prepared as a precipitation agent. 90 ml of a saturated sodium carbonate solution was added into a 0.5 L batch reactor containing 300 mL of lithium and magnesium sulfate solution under a feeding rate of $3 mL min^{-1}$. A reaction was conducted at 90 °C for 1 h. Precipitated $Li_2CO_3$ powder was gathered by filtration and then washed by boiling deionized water. The washed powders were dried and carefully ground in a mortar.

$Ni_{0.6}Co_{0.2}Mn_{0.2}(OH)_2$ hydroxide precursors (Bare precursor) were prepared via a typical coprecipitation method. The 1 L of 1.0 M $NH_4OH$ solution was purged under $N_2$ before reaction to remove other residual reactive gas species. 2.0 M aqueous solution of mixed $NiSO_4 \cdot 6H_2O$, $CoSO_4 \cdot 7H_2O$, and $MnSO_4 \cdot H_2O$ with a Ni:Co:Mn molar ratio of 6:2:2 was pumped into the reactor under $N_2$ atmosphere. For Mg 1.5% and 3.5% precursor, certain amounts of $MgSO_4$ were added into the mixture solution. Concurrently, 5.0 M of NaOH solution and the desired amounts of $NH_4OH$ solution as a chelating agent were separately pumped into the reactor. The pH value in the reactor was maintained 10.5 at 50 °C with a stirring speed of 1500 r.p.m. with an overhead stirrer. The obtained hydroxide precursors were filtered, washed with deionized water, and dried in a vacuum oven overnight.

Control $LiNi_{0.6}Co_{0.2}Mn_{0.2}O_2$ was prepared by thoroughly mixing the Bare precursor with a stoichiometric amount of LCB. In cases of LCD and LCD-HC, prepared LCM and LCM-HC were mixed with Bare precursor, respectively. Likewise, SSD and SSD-HC powder were prepared with stoichiometric amounts of $MgCO_3$ hydrate with a Bare precursor and LCB mixture. For CPD and CPD-HC, Mg 1.5 and 3.5% precursor were mixed with LCB instead of Bare precursor, respectively. Li/TM molar ratios of all mixtures were fixed with a 1.05 ratio to compensate for lithium loss under high-temperature conditions. All mixtures were calcined at 850 °C for 12 h in a tube furnace under $O_2$ gas flowing.

For better understanding, a visual summary of synthesized materials is provided in Supplementary Note S1.

### Electrochemical testing

The obtained active materials were mixed with a carbon black and polyvinylidene fluoride (PVDF) in a weight ratio of 90:5:5 (active material:carbon black:PVDF) in *N*-methyl-2-pyrrolidone (NMP). The slurry was cast onto the Al foil current collector ($5-6 mg cm^{-2}$ loading of active materials). The electrochemical performance of half cells was evaluated using 2032 coin cells with a Li metal (counter/reference electrode) and 1 M $LiPF_6$ in EC:DEC (50:50 vol%, electrolyte). The Li metal electrode was prepared by punching the Li metal foil into a disc shape. The half cells were then cycled between 3.0 and 4.3 V (vs. $Li^+$/Li)

using WBCS3000 (WonATech) in a temperature-controlled chamber. Every cell was rested over 10 h before cycling and pre-cycled twice.

The electrochemical performances of full cells were evaluated using 2032 coin cells with a graphite anode and 1.3 M $LiPF_6$ in EC:DEC (30:70 vol%, electrolyte). The cathode and anode (graphite anode powder, Sigma-Aldrich) active materials were loaded 13-14 and $7-8 mg cm^{-2}$, respectively onto the Al and Cu foil current collector, respectively. The graphite anode was mixed with a carbon black, styrene-butadiene rubber (SBR), and carboxymethyl cellulose (CMC) in a weight ratio of 94.5:1.3:2.8:1.4 (active material:carbon black:SBR:CMC) in deionized water. The N/P ratio and the overhang fraction were controlled to -1.1 and 12.9%, respectively. After cell assembly, the full cells were wetted at 1.5 V for 12 h and formation cycle was conducted between 3.0 and 4.2 V (vs. graphite) using WBCS3000 (WonATech) in a temperature-controlled chamber at 40 °C. The optimization of full cells was based on the recent report by Garayt et al.[36].

### XRD and Rietveld refinement

XRD spectra of lithium carbonates were obtained on the PLS-II XRS-GIST 5D beamline at the Pohang Accelerator Laboratory (PAL) with a wavelength of 1.24007 Å. The XRD (D8 ADVANCES, Bruker) spectra of precursors and cathode materials were measured with a Cu Kα wavelength of 1.5418 Å. Rietveld refinements of the collected XRD patterns were carried out using FullProf Suite package[37].

### Neutron diffraction and Rietveld refinement

ND data of $LiNi_{0.6}Co_{0.2}Mn_{0.2}O_2$ was obtained from HANARO facility at the Korea Atomic Energy Research Institute (KAERI). The measurement was conducted in the 2θ range of 10–160° with a step size of 0.05° using a constant wavelength of 1.834583 Å. Rietveld refinements quantitative atomic analysis of the collected ND patterns were also carried out using FullProf Suite package[37].

### Neutron diffraction pattern simulation

ND pattern simulation was conducted by Pattern calculation (Neutron −CW) in FullProf Suite package[37]. The simulated patterns depended on Mg doping site were acquired by adjusting occupancy parameters based on Rietveld parameters of pure $LiNi_{0.6}Co_{0.2}Mn_{0.2}O_2$.

### X-ray absorption

XAS spectra of cathode materials were obtained on the PLS-II 10 C Wide XAFS beamline at the Pohang Accelerator Laboratory (PAL). All powder samples were analysed between Kapton tape with appropriate thickness for transmission mode detection. Spectra fitting was conducted by ATHENA software package[38] and all spectra were aligned by reference foil.

### X-ray photoelectron spectroscopy

XPS spectra of lithium carbonates were obtained on the PLS-II 10A2 HR-PES II beamline at the Pohang Accelerator Laboratory (PAL). Mg 2*p* XPS spectra were obtained at 961.06 eV with a pass energy of 50 eV. The spectra were obtained under ultra-high vacuum (-$10^{-9}$ Torr) after aligning Au 4*f* reference peak position.

### pH meter

pH meter (913 pH Meter, Metrohm) was used to trace the pH value in the lithium carbonate synthesis reactor. pH meter (916 Ti-Touch, Metrohm) and dispenser (800 Dosino, Metrohm) were used to maintain the pH value in the hydroxide precursor synthesis reactor. The pH meters were calibrated before measurement by three standard solutions.

### ICP-OES

The chemical compositions of the lithium carbonates, hydroxide precursors, and cathode materials were determined by an ICP-OES (PerkinElmer Optima 5300 DV, PerkinElmer, Inc.). Lithium carbonates

were dissolved in an HCl solution. Hydroxide precursors and cathode materials were dissolved in *aqua regia* (HCl:HNO$_3$ = 3:1). All dissolved samples were diluted in deionized water. Calibration curves were generated using at least three standard solutions, with the results used only from correlation coefficients that were greater than 0.999 and relative standard deviation (RSD) <5%.

## Scanning electron microscopy

High-resolution SEM-EDS images were obtained by FE-SEM (JSM 7800 F PRIME with Dual EDS, JEOL) and EDS (Aztec, Oxford) with an accelerating voltage of 15 kV. SEM analyses of hydroxide precursors and cathode materials were carried out on a TE-SEM (Genesis-1000, EMCRAFT) with an accelerating voltage of 10 kV. Cross-section SEM-EDS images were obtained by FE-SEM (JSM 7100F, JEOL) and EDS (Aztec, Oxford) with an accelerating voltage of 15 kV. The Li$_2$CO$_3$ powder sample was casted on Al foil with PVDF binder and polished by a cross-section polisher (SM-09010, JEOL).

## Intermediated mixture analysis

The intermediated mixture analysis was conducted. The intermediated mixture sample of LCD was prepared by mixing Bare precursor with LCM lithium precursor. Likewise, the intermediated mixture sample of SSD was prepared by mixing Bare precursor with LCB lithium precursor with a stoichiometric amount of MgCO$_3$. All the mixtures were then calcinated at 400 °C for 5 h in a tube furnace under O$_2$ gas flowing. The chemical composition analysis was conducted by SEM-EDS. SEM backscattered electron images and EDS line scan images were obtained by FE-SEM (JSM-7100F) and EDS (Aztec, Oxford) with an accelerating voltage of 15 kV. SEM-EDS mapping images were obtained by FE-SEM (JSM 7800 F PRIME with Dual EDS, JEOL) and EDS (Aztec, Oxford) with an accelerating voltage of 15 kV. The crystal structure analysis was conducted by XRD (D8 ADVANCES, Bruker) with a Cu Kα wavelength of 1.5418 Å.

## Lithium carbonate clogging test

Lithium carbonate clogging test was conducted in 2 L batch reactor with overhead stirrer. 1.2 L of aqueous solution was prepared in the reactor with a concentration of 10 g L$^{-1}$ of Li$^+$. In the case of the reaction with residual Mg impurity, 0.3 g L$^{-1}$ of Mg$^{2+}$ was added. Both tests were conducted in the same reactor and stirrer. Sodium carbonate saturated solution was pumped into the reactor under a feeding rate of 12 ml min$^{-1}$ for 30 min. After that, an additional reaction for 30 min was conducted to meet equilibrium maintaining 90 °C and a stirring speed of 400 r.p.m. with an overhead stirrer. The reacted batch was rested for a certain period to settle Li$_2$CO$_3$ powder down. At the same resting time, the status of both reactor batches was observed.

## Pilot plant scale lithium carbonate extraction test

Through the pilot plant scale (annual plant production capacity 2500 tons of Li$_2$CO$_3$ per year) lithium carbonate manufacturing test, the behaviour of lithium and impurities in each stage were tested. Brine LX P/P test was conducted using actual brine pumped from 'Hombre Muerto' Salt Lake in the Argentine highlands and the hard rock LX P/P test was conducted using spodumene mined from Western Australia and concentrated through the following pretreatment processes. The concentrations of all samples were analysed by ICP-OES (SEPCTRO ARCOS, AMETEK, Inc.). In particular, in the case of the Mg refinery test, the industrial grade of Ca(OH)$_2$ (85% purity) was used as a reagent reflecting the conditions of the actual manufacturing process.

## Particle size distribution analysis

Ex-situ particle size distribution (PSD) analysis of LCM and LCB was carried out using a particle size analyser (Mastersizer 3000, Malvern). The synthetic conditions for LCM and LCB were identical to those described in the "Material synthesis" section, with the only variation

being the reaction time. The PSD analysis result of Mg(OH)$_2$ byproduct was obtained by particle size analyser (Mastersizer 3000, Malvern).

## Economic and environmental analysis

The results of economic evaluation and CO$_2$ emission presented in this work are based on the results of actual brine and hard rock LX processes (annual plant production capacity 25,000 tons of Li$_2$CO$_3$ per year). Raw materials and reagents input during the process were calculated by applying consumption data from the actual plant. In addition, the conditions of the LX process reflect the operating conditions in the Argentine Salt Lake. The processing cost and economic feasibility were evaluated using the mass balance and energy balance derived from the actual plant. The calculation of CO$_2$ emission is based on this result as well.

## Data availability

All data analysed and generated during this study are included in the article and its Supplementary Information. Source data are provided with this paper.

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

## Acknowledgements

This work was supported by the grant of National Research Foundation of Korea, South Korea (2019M3D1A1079306–Y.-T.K., 2022R1F1A1063367–K.-Y.P., and 2022R1A2C3007855–Y.-T.K.), Korea Institute of Energy Technology Evaluation and Planning (20217510100020–Y.-T.K.), and Korea Institute for Advancement of Technology (KIAT) grant funded by the Korea Government (MOTIE) (P0012748–K.-Y.P., HRD Programme for Industrial Innovation).

## Author contributions

G.C. conceived the idea and designed the experiments. G.C. synthesized the samples and did the electrochemical, XRD, Rietveld refinement of XRD and ND, XAS, XPS, pH meter, ICP-OES, SEM, intermediate mixture analysis, and LC clogging test. H.K. operated ND analyses. J.K. and G.C. did cross-sectional SEM analyses. G.C. and K.-Y.P. did ND diffraction pattern simulation analyses. W.J. did pilot plant scale lithium extraction test, particle size distribution analyses, and economic and environmental analyses. G.C., K.-Y.P. and Y.-T.K. discussed the results and wrote the paper with contributions from all authors.

## Competing interests

The authors declare no competing interests.
