## [Peer Review File · Nature Communications]

REVIEWER COMMENTS

Reviewer #1 (Remarks to the Author):

Choe et al studied the effect of Mg impurity (doping) on the process properties (quality, yield, cost, potential CO₂ footprint) for Li₂CO₃ extraction, as well as its impact on the synthesis of NCM622 cathodes. Crystal morphology, microstructure and electrochemical performance were all investigated. The authors claimed clear advantages of producing such Mg-doped Li₂CO₃ in all aspects. This is an interesting study, but a number of issues need to be addressed/discussed:

- 1) The authors only discussed the impurity effect of Mg, but how about other cations such as Al?
- 2) The effect of Mg in the Li-containing solution towards precipitation and less likely on the clogging is not very clear. A more direct study on the nucleation and growth process might be necessary to confirm the claims and provide more insights. A study on the Mg distribution inside Li₂CO₃ particles will be very useful to describe the grow process.
- 3) The authors claim that Mg removal is a major cost-driver in Li₂CO₃ production, how this conclusion is made? It will be useful to provide specific numbers in cost and CO₂ footprint associated with each step in the process flowchat, if possible or at least those key steps. There are a few similar/comparable unit operation that are supposed to induce major cost as well.
- 4) The authors only concern the Mg effect on Li₂CO₃ production, but LiOH is more commonly used for making high-Ni cathodes so it will be interesting to study its effect on LH, as well as its influence in high Ni cathodes.
- 5) MgCO₃ has lower point than Li₂CO₃, so phase separation will happen anyway during sintering process as temperature ramps. Therefore, it will be confusing to see the difference from a Mg-doped Li₂CO₃ (LCD) to solid-state synthesis with separate carbonates (SSD) when they are sintered with NCM622 precursors? How the structure/composition difference happens
- 6) How does the Mg-doping in different sites affect the ICE? Some explanation is necessary
- 7) the CAPEX and OPEX modeling are a little too simple. The authors need to provide more items cost and associated CO₂ footprint. With the current general parameters, it is hard to evaluate the quality of such modeling results.
- 8) There are also a number for spelling/grammar issues in the main text and SI. The authors need to proof check carefully and revise accordingly.

Reviewer #2 (Remarks to the Author):

In this manuscript, the authors proposed that low-grade lithium carbonate may be a cost-effective option for the battery cathode materials, using the example of Mg impurity. This manuscript provides an interesting new perspective and some interesting results on Li_2CO_3 growth kinetics and NCM doping. However, I am struggling to suggest that it may be accepted in the current form at Nature Commun.. Perhaps a significantly revised version may be considered.

1) Battery-grade Li_2CO_3 does not just mean high-purity, it also means (or implies) high consistency. The final product (such as NCM622 shown in this work) may exhibit a large variation in the accessible Li-storage capacity and cycle stability when the Mg amount varies (shown by the authors in Supplementary Fig. 11). This is not acceptable in the battery industry. I am wondering how it is possible to perfectly control the amount of Mg impurity in the low-grade Li_2CO_3 , especially when this production process is scaled up to a practically-relevant scale. Would it be easier to first make the battery-grade Li_2CO_3 and then add the Mg-containing dopants? In a lot of cases, other dopants (such as Zr, W, and Mo) will be added during the sintering process anyway (because they are not in the Brine or the hard-rock). Having a Mg-containing lithium precursor does not seem to simplify the cathode material production process.

2) Although the 1.5 mol% LCD Mg-doped NCM622 showed better electrochemical performance than the SSD and CPD samples. The improvement was not significant and only one cell was shown for each case. Results from multiple cells and full-cell tests should be provided for a paper considered for Nature Commun. The doping amount is also quite high. The total doping concentration for industrial production is usually below 5000 ppm (~0.5 wt%). Doping usually reduces the specific capacity for the NCM cathodes. It is a bit strange to see the bare sample showed the lowest capacity in Figure 3a and 3b. This makes me wonder if the authors prepared their samples in the right way. If the authors check the literature, they would realize NCM622 are usually prepared at 900 degree C in air.

3) NCM622 is more like a "transition" material for the battery industry. For high-Ni cathodes, $\text{LiOH}\cdot\text{H}_2\text{O}$ is required. What is the challenge for making Mg-containing low-grade $\text{LiOH}\cdot\text{H}_2\text{O}$?

Reviewer #3 (Remarks to the Author):

This work discussed the roles of certain impurities such as Mg^{2+} in raw lithium salt and its impact in the as-prepared NMC cathode materials. It tackles the purity and cost issue of raw materials from a different aspect which is of great important to industry. The upper limit of 1%Mg in "not-that-pure" Li_2CO_3 largely relied industry cost in refining brines/rocks when extracting Li_2CO_3 . The authors further explored the doping site of Mg provided from three different approaches which is also quite interesting. The techno-economic impacts of this work is well discussed and provides insights for fundament research.

Some questions and suggestions are listed below.

1. The authors mentioned that growth of Li_2CO_3 into octahedrons is probably resulted from the cubic morphology of magnetite seeds. Do they have evidence for that? SEM images of magnetite for example?
2. Morphology not only impacts the flowability and compressibility drying powder production, they also change the "filling ratio" of the furnace when synthesizing NMC which is determined by the density of salts. Is there any difference for Li_2CO_3 with different contents of Mg impurities and morphologies?
3. As the electrochemical performances are conducted in half cells the variation of NMC developed by using different Li_2CO_3 salts are not convincing. Please add some discussion in the paragraph to clarify that in the future full coin cells are still needed to assess the usable capacities and cycling stability. If there are more Mg occupying Li sites, then the usable capacity will be lower in those NMC which, however, cannot be reflected in half cells.
4. The roles and impacts of impurities in raw salts was discussed in a recent perspective (Nature Energy, 2023,329-339. Please cite this relevant work.

Response to Reviewers

REVIEWER COMMENTS

Reviewer #1 (Remarks to the Author):

Choe et al studied the effect of Mg impurity (doping) on the process properties (quality, yield, cost, potential CO₂ footprint) for Li₂CO₃ extraction, as well as its impact on the synthesis of NCM622 cathodes. Crystal morphology, microstructure and electrochemical performance were all investigated. The authors claimed clear advantages of producing such Mg-doped Li₂CO₃ in all aspects. This is an interesting study, but a number of issues need to be addressed/discussed:

- 1) The authors only discussed the impurity effect of Mg, but how about other cations such as Al?
- 2) The effect of Mg in the Li-containing solution towards precipitation and less likely on the clogging is not very clear. A more direct study on the nucleation and growth process might be necessary to confirm the claims and provide more insights. A study on the Mg distribution inside Li₂CO₃ particles will be very useful to describe the grow process.
- 3) The authors claim that Mg removal is a major cost-driver in Li₂CO₃ production, how this conclusion is made? It will be useful to provide specific numbers in cost and CO₂ footprint associated with each step in the process flowchart, if possible or at least those key steps. There are a few similar/comparable unit operation that are supposed to induce major cost as well.
- 4) The authors only concern the Mg effect on Li₂CO₃ production, but LiOH is more commonly used for making high-Ni cathodes so it will be interesting to study its effect on LH, as well as its influence in high Ni cathodes.
- 5) MgCO₃ has lower point than Li₂CO₃, so phase separation will happen anyway during sintering process as temperature ramps. Therefore, it will be confusing to see the difference from a Mg-doped Li₂CO₃ (LCD) to solid-state synthesis with separate carbonates (SSD) when they are sintered with NCM622 precursors? How the structure/composition difference happens
- 6) How does the Mg-doping in different sites affect the ICE? Some explanation is necessary
- 7) the CAPEX and OPEX modeling are a little too simple. The authors need to provide more items cost and associated CO₂ footprint. With the current general parameters, it is hard to evaluate the quality of such modeling results.
- 8) There are also a number for spelling/grammar issues in the main text and SI. The authors need to proof check carefully and revise accordingly.

Reviewer #2 (Remarks to the Author):

In this manuscript, the authors proposed that low-grade lithium carbonate may be a cost-effective option for the battery cathode materials, using the example of Mg impurity. This manuscript provides an interesting new perspective and some interesting results on Li_2CO_3 growth kinetics and NCM doping. However, I am struggling to suggest that it may be accepted in the current form at Nature Commun.. Perhaps a significantly revised version may be considered.

1) Battery-grade Li_2CO_3 does not just mean high-purity, it also means (or implies) high consistency. The final product (such NCM622 shown in this work) may exhibit a large variation in the accessible Li-storage capacity and cycle stability when the Mg amount varies (shown by the authors in Supplementary Fig. 11). This is not acceptable in the battery industry. I am wondering how it is possible to perfectly control the amount of Mg impurity in the low-grade Li_2CO_3 , especially when this production process is scaled up to a practically-relevant scale. Would it be easier to first make the battery-grade Li_2CO_3 and then add the Mg-containing dopants? In a lot cases, other dopants (such Zr, W, and Mo) will be added during the sintering process anyway (because they are not in the Brine or the hard-rock). Having a Mg-containing lithium precursor does not seem to simplify the cathode material production process.

2) Although the 1.5 mol% LCD Mg-doped NCM622 showed better electrochemical performance than the SSD and CPD samples. The improvement was not significant and only one cell was shown for each case. Results from multiple cells and full-cell tests should be provided for a paper considered for Nat. Commun. The doping amount is also quite high. The total doping concentration for industrial production is usually below 5000 ppm (~0.5 wt%). Doping usually reduce the specific capacity for the NCM cathodes. It is a bit strange to see the bare sample showed the lowest capacity in Figure 3a and 3b. This makes me wonder if the authors prepared their samples in the right way. If the authors check the literature, they would realize NCM622 are usually prepared at 900 degree C in air.

3) NCM622 is more like a "transition" material for the battery industry. For high-Ni cathodes, $\text{LiOH}\cdot\text{H}_2\text{O}$ is required. What is the challenge for making Mg-containing low-grade $\text{LiOH}\cdot\text{H}_2\text{O}$?

Reviewer #3 (Remarks to the Author):

This work discussed the roles of certain impurities such as Mg^{2+} in raw lithium salt and its impact in the as-prepared NMC cathode materials. It tackles the purity and cost issue of raw materials from a different aspect which is of great important to industry. The upper limit of 1%Mg in "not-that-pure" Li_2CO_3 largely relied industry cost in refining brines/rocks when extracting Li_2CO_3 . The authors further explored the doping site of Mg provided from three different approaches which is also quite interesting. The techno-economic impacts of this work is well discussed and provides insights for fundament research.

Some questions and suggestions are listed below.

1. The authors mentioned that growth of Li_2CO_3 into octahedrons is probably resulted from the cubic morphology of magnetite seeds. Do they have evidence for that? SEM images of magnetite for example?
2. Morphology not only impacts the flowability and compressibility drying powder production, they also change the "filling ratio" of the furnace when synthesizing NMC which is determined by the density of salts. Is there any difference for Li_2CO_3 with different contents of Mg impurities and morphologies?
3. As the electrochemical performances are conducted in half cells the variation of NMC developed by using different Li_2CO_3 salts are not convincing. Please add some discussion in the paragraph to clarify that in the future full coin cells are still needed to assess the usable capacities and cycling stability. If there are more Mg occupying Li sites, then the usable capacity will be lower in those NMC which, however, cannot be reflected in half cells.
4. The roles and impacts of impurities in raw salts was discussed in a recent perspective (Nature Energy, 2023,329-339. Please cite this relevant work.

Editorial Note: The figures shown on next page are reproduced from K. Sole et al Flowsheet Options for Cobalt Recovery in African Copper–cobalt Hydrometallurgy Circuits (2018) from the ALTA Free Library, with express permission from ALTA Metallurgical Services.

Reply to Reviewer #1

Choe et al studied the effect of Mg impurity (doping) on the process properties (quality, yield, cost, potential CO₂ footprint) for Li₂CO₃ extraction, as well as its impact on the synthesis of NCM622 cathodes. Crystal morphology, microstructure and electrochemical performance were all investigated. The authors claimed clear advantages of producing such Mg-doped Li₂CO₃ in all aspects. This is an interesting study, but a number of issues need to be addressed/discussed:

Response: We thank very much for your interest in our work and the kind support for the publication of our work.

1) The authors only discussed the impurity effect of Mg, but how about other cations such as Al?

Response: We appreciate the reviewer's insightful comment. In the case of hard rock lithium production, spodumene ores often contain significant levels of Al impurities. However, it is worth emphasizing that the Al element readily precipitates as a hydroxide under pH 7 conditions (see Figure below). As a result, during the Mg elimination step, where we raise the pH using a Ca(OH)₂ reagent (beyond pH 7), the Al impurity is completely removed. Therefore, the Al element is automatically eliminated following the Mg elimination step.

Additionally, it is worth noting that Al or other dopant impurities may have the potential to positively influence the cathode material's performance. However, in the context of the elimination process, accommodating these dopants could result in a significant remaining Mg impurity, which might introduce challenges in achieving the desired level of cathode material properties.

Figure. Solubility of base metal hydroxides (left) and carbonates (right) as a function of pH (retrieved from Sole, Kathryn & Parker, John & Cole, Peter & Mooiman, Michael. (2018). Flowsheet options for cobalt recovery in African copper-cobalt hydrometallurgy circuits.).

2) The effect of Mg in the Li-containing solution towards precipitation and less likely on the clogging is not very clear. A more direct study on the nucleation and growth process might be necessary to confirm the claims and provide more insights. A study on the Mg distribution inside Li_2CO_3 particles will be very useful to describe the grow process.

Response: To investigate nucleation and growth behavior with and without Mg impurity, we compared changes in particle size distribution (PSD) for LCM and LCB over time. In the initial stages of particle growth (as shown in the Figure below, 5 min), Mg-containing LCM exhibited a higher population of smaller-sized particles due to the presence of $\text{MgCO}_3 \cdot x\text{H}_2\text{O}$ seeds, in contrast to LCB without Mg impurity. As the reaction time increased, the PSD of LCM became significantly narrower than that of LCB, demonstrating the typical growth behavior when there are more nuclei mediating the process. We have included this ex-situ PSD analysis as **Supplementary Fig. S7** to illustrate the direct seeding effect on particle growth, with a detailed explanation provided in the **Methods** section.

Furthermore, we confirmed the distribution of Mg within LCM-HC particles through cross-section SEM-EDS analysis. This analysis revealed a dense cross-section of particles, supporting the single-crystal-like morphology of Mg-containing Li_2CO_3 particles. Mg^{2+} was found to be homogeneously distributed inside the particles, consistent with solid solution behavior. We have included this study as **Supplementary Fig. S9** to provide further evidence of the single-crystal-like morphology and solid solution behavior.

These additional experimental results, as demonstrated in **Supplementary Figure 10**, are expected to aid in suppressing the clogging phenomenon during the precipitation reaction.

Figure. Ex-situ PSD analysis was conducted based on the reaction time of LCM (left) and LCB (right). Smaller-sized particles were more prevalent in the early-stage reaction (~5 min), which is believed to be due to the presence of $\text{MgCO}_3 \cdot x\text{H}_2\text{O}$ nanoparticle species. Subsequently, a narrower PSD was observed at the reaction batch containing Mg impurities.

Figure. Cross-section SEM-EDS mapping of LCM-HC. Scale bars, 5 μm . Single-crystal-like morphology was observed without pore and Mg^{2+} was evenly distributed within Li_2CO_3 particles, consistent with solid solution behavior. The LCM-HC powder sample was casted on Al foil with PVDF binder and polished by cross-section polisher.

3) The authors claim that Mg removal is a major cost-driver in Li_2CO_3 production, how this conclusion is made? It will be useful to provide specific numbers in cost and CO_2 footprint associated with each step in the process flowchart, if possible or at least those key steps. There are a few similar/comparable unit operation that are supposed to induce major cost as well.

Response: Thank you for this comment. The cost of filter press facilities constitutes the majority of the refinery cost. These facilities are used for both Mg and Ca elimination processes. As mentioned in our manuscript, Mg elimination accounts for more than two-thirds of the overall refinery costs compared to Ca refinery (**Fig. 1d**). This cost disparity primarily arises from the smaller precipitate particles and higher impurity concentrations in Mg refining. The ultrafine $\text{Mg}(\text{OH})_2$ precipitate from Mg refinery requires 2–3 times the processing time compared to CaCO_3 , which leads to a proportional increase in the size of the facilities, ultimately resulting in higher CAPEX and OPEX. We kindly ask the reviewer to understand that specific cost figures are hard to be disclosed due to the

sensitivity of the information as corporate secrets of our partner company, which could potentially reveal our process design. Instead, we have provided the CO₂ footprint associated with each step in the process flowchart (see **Supplementary Figs. S1 and S2**).

4) The authors only concern the Mg effect on Li₂CO₃ production, but LiOH is more commonly used for making high-Ni cathodes so it will be interesting to study its effect on LH, as well as its influence in high Ni cathodes.

Response: We appreciate the reviewer for bringing up this point. In the original manuscript, we investigate the impact of Mg on Li₂CO₃ production due to its priority as a strategic and universal material. Indeed, the issue of Mg impurities is more pronounced in LH production compared to LC. Impurity elements such as Na⁺ and Mg²⁺ cannot be easily washed out in LH production because LH is also soluble in water. Moreover, a significant loss of LH occurs in the repeated washing process. Thus, the economic and environmental benefit would be greater in LH production when considering these concerns.

[Redacted]

5) MgCO₃ has lower point than Li₂CO₃, so phase separation will happen anyway during sintering process as temperature ramps. Therefore, it will be confusing to see the difference from a Mg-doped Li₂CO₃ (LCD) to solid-state synthesis with separate carbonates (SSD) when they are sintered with NCM622 precursors? How the structure/composition difference happens

Response: Thanks for this comment. We have demonstrated the solid solution behavior of Mg-containing Li₂CO₃ by utilizing synchrotron-sourced XRD, varying the concentration of Mg (please see **Fig. 2a–d and Supplementary Note S2** in the revised manuscript). Additionally, we characterized the intermediate state using XRD (**Supplementary Fig. S34**) and SEM-EDS (**Supplementary Figs. S31, S32, and S33**) to illustrate the kinetical structure/composition differences between LCD and SSD during the cathode synthetic process.

Supplementary Fig. S31. SEM-EDS line scan images of LCD and SSD mixtures. In LCD mixture, high Ni, Co, and Mn signal was detected at transition metal oxide particles, but negligible Mg signal. However, high Mg signal was detected at Li_2CO_3 particle that Mg impurity containing Li_2CO_3 (LCM) still hold Mg element inside. On the other hand, in SSD mixture, high Mg signal was detected at transition metal oxide particles with high Ni, Co, and Mn signal. The MgCO_3 in SSD mixture was reacted with transition metal oxide producing (Ni, Co, Mn, Mg)-O. Scale bars; 10 μm .

Supplementary Fig. S34. XRD analysis of intermediate phases. (A) Ex-situ XRD analysis of Bare by temperature. (B) XRD patterns of intermediate LCD and SSD mixtures calculated at 400 °C for 5 h. (C) Overlaid XRD patterns with peak indices. The peak assignments of ()_L, ()_{RS}, and * refer to the layered, rocksalt, and lithium carbonate structures, respectively.

The formation of a $\text{MgCO}_3\text{-Li}_2\text{CO}_3$ solid solution was suggested by the gradual high-angle shift of the Bragg peaks (Fig. 2a–d), and the concentration of Mg was confirmed to be 0.98 wt% using ICP-OES (Supplementary Table S3). In typical phase diagrams of binary solid solution systems, the melting point of the solid solution is primarily influenced by the major component. Therefore, an increase in the melting point of the minor phase MgCO_3 in the binary phase diagram is justifiable. Due to these reasons, Mg^{2+} in LCM was able to co-diffuse with Li^+ simultaneously during sintering at higher temperature compared to MgCO_3 alone. To clarify the concept of solid solution, we added figure below as Supplementary Fig. S35 and mentioned at our revised manuscript page 12, line number 19.

Figure. The conceptual phase diagram of solid solution LCM. The dilute MgCO₃ in Li₂CO₃ solid solution is expected to closely follow the melting behavior of the primary phase of Li₂CO₃.

6) How does the Mg-doping in different sites affect the ICE? Some explanation is necessary.

Response: We thank the reviewer for raising this point. We re-evaluated the charge/discharge curves of the first cycle for the cathodes under different cell compositions and test conditions. The cathodes demonstrated similar initial coulombic efficiency (ICE) at approximately 91% and 93% for 25 °C and 40 °C half-cell tests, respectively, and approximately 89% for the full-cell test. We did not observe significant differences of ICE among the Mg-doped cathodes; however, there was a slight increase in LCD. This suggests that the higher Mg²⁺ site-selectivity and decreased cation mixing ratio led to an increase in the *c* lattice parameter, enhancing Li⁺ diffusion kinetics (*Phys. Chem. Chem. Phys.*, 2016, 18, 3956-3965). We additionally mentioned this discussion in our **revised SI, Supplementary Text for Supplementary Fig. S26** as follows:

*“The improvement in rate capability and the slight increase in initial coulombic efficiency (ICE) in LCD (Fig. 3) are suggested to be a result of the increased *c* lattice parameter, which enhances Li⁺ diffusion kinetics^{S5}.”*

7) the CAPEX and OPEX modeling are a little too simple. The authors need to provide more items cost and associated CO₂ footprint. With the current general parameters, it is hard to evaluate the quality of such modeling results.

Response: We appreciate the reviewer's comment. We have included the details of the CO₂ footprint at each

process step in **Supplementary Figs. S1 and S2**. However, as previously mentioned in our *response to comment number 3*, the evaluation models are based on pilot plant test results, which are closely linked to our actual working plant of industrial partner. Hence, it is challenging to provide the specific details the reviewer may have anticipated. However, we wish to reassure the reviewer that our evaluations are based on well-founded models and calculations, relying on the results provided by our trusted partner engineering design companies (Hatch Ltd. and Veolia) for the development of commercial facilities.

8) There are also a number for spelling/grammar issues in the main text and SI. The authors need to proof check carefully and revise accordingly.

Response: We appreciate the reviewer for pointing this out. We carefully proofread and made appropriate revisions.

Reply to Reviewer #2

In this manuscript, the authors proposed that low-grade lithium carbonate may be a cost-effective option for the battery cathode materials, using the example of Mg impurity. This manuscript provides an interesting new perspective and some interesting results on Li_2CO_3 growth kinetics and NCM doping. However, I am struggling to suggest that it may be accepted in the current form at Nature Commun.. Perhaps a significantly revised version may be considered.

Response: We thank the reviewer for appreciating the important results in our manuscript and comments. We have carefully considered each of the reviewer's concerns, and we address them one-by-one below.

1) Battery-grade Li_2CO_3 does not just mean high-purity, it also means (or implies) high consistency. The final product (such NCM622 shown in this work) may exhibit a large variation in the accessible Li-storage capacity and cycle stability when the Mg amount varies (shown by the authors in Supplementary Fig. 11). This is not acceptable in the battery industry. I am wondering how it is possible to perfectly control the amount of Mg impurity in the low-grade Li_2CO_3 , especially when this production process is scaled up to a practically-relevant scale. Would it be easier to first make the battery-grade Li_2CO_3 and then add the Mg-containing dopants? In a lot cases, other dopants (such Zr, W, and Mo) will be added during the sintering process anyway (because they are not in the Brine or the hard-rock). Having a Mg-containing lithium precursor does not seem to simplify the cathode material production process.

Response: We are pleased to provide a more detailed explanation of the lithium extraction process in terms of consistency. Indeed, the high consistency of Mg impurity concentration in Li_2CO_3 can be achieved by simply adjusting the refining process, which involves controlling the amount of reagent and optimizing S/L filtration facilities on a practically-relevant scale. The concentration of Mg can be controlled by adjusting the input of $\text{Ca}(\text{OH})_2$ and the number of S/L filtration steps, ensuring consistent impurity concentration. In addition, the ability to control specific concentrations of certain elements is a fundamental capability in every mass production chemical process and depends on the controllability of the factory.

We would like to provide additional explanations regarding **Supplementary Note S4** below:

The **second inset table in Supplementary Note S4** demonstrates that regardless of the initial Mg content in feed

leachate, the consistency of Mg concentration at post-refinery solution was maintained to $\sim 0.1 \text{ g L}^{-1}$. This was achieved by adjusting the input of Ca(OH)_2 and the S/L filtration steps for the Mg(OH)_2 byproduct. This strategy remains consistent when applied at a practically-relevant scale, with the optimization of relevant variables.

Table. Acid roasting impurity test. The composition of Li_2SO_4 leachate and solution after Mg elimination step by H_2SO_4 input.

H_2SO_4 input (g)	Sample	Li (g L^{-1})	S (g L^{-1})	Mg (g L^{-1})	K (g L^{-1})
20	Leachate	11.01	31.14	0.268	0.026
	Mg elimination step	11.07	25.04	0.102	0.021
40	Leachate	11.77	28.53	3.16	0.033
	Mg elimination step	11.24	22.71	0.127	0.031
80	Leachate	11.72	25.45	4.97	0.058
	Mg elimination step	11.31	24.81	0.108	0.052

We had addressed it in **Supplementary Note S4**, as mentioned in our **revised manuscript on page 9, line number 11–12**:

*“Furthermore, the level of Mg impurity in the lithium precursor can easily be regulated by adjusting the refining process (see detailed explanation in **Supplementary Note S4**).”*

The solid-state doping is one of the simpler methods for doping cathode materials; however, it is hard to find a reason not to utilize Mg-containing Li_2CO_3 with its advantages in kinetical site-selectivity and economic/environmental benefits. Other dopants such as Zr, W, and Mo would be compatible with LCM and pose no issues in the conventional doping process.

2) Although the 1.5 mol% LCD Mg-doped NCM622 showed better electrochemical performance than the SSD and CPD samples. The improvement was not significant and only one cell was shown for each case. Results from multiple cells and full-cell tests should be provided for a paper considered for Nat. Commun. The doping amount is also quite high. The total doping concentration for industrial production is usually below 5000 ppm ($\sim 0.5 \text{ wt}\%$). Doping usually reduce the specific capacity for the NCM cathodes. It is a bit strange to see the bare sample showed

the lowest capacity in Figure 3a and 3b. This makes me wonder if the authors prepared their samples in the right way. If the authors check the literature, they would realize NCM622 are usually prepared at 900 degree C in air.

Response: We thank for this critical review. To address this concern, we performed a comprehensive re-evaluation of the electrochemical performance of the cathodes by retesting both half cells and additional full cells at least three times. We acknowledge that the initial capacity test was conducted at a high current density and a low temperature, which could have potentially influenced the initial capacity properties of the cathodes. This suggests that the rate capability properties (**Fig. 3g**) may indeed be intertwined with the initial capacity properties.

We re-evaluated the charge/discharge curves of every cathode material presented in this work at a current rate of 16 mA g⁻¹ with 25 °C. The revised charge/discharge curves are used in **Fig. 3** and **Supplementary Figs. S14 and S19**. In this case, the Bare cathode exhibited the highest initial capacity, possibly due to the redox-inactive nature of Mg²⁺. Furthermore, to support and verify the enhanced performance, we conducted additional high temperature cycling tests and full cell tests (**Fig. 3**). This allowed us to demonstrate the significance of the improvements and further validate the enhanced cyclability observed in the LCD cathode. The consistency in performance (initial capacity and cyclability) was evident not only at 25 °C but also at elevated temperatures of 40 °C and in full cell configurations.

We have revised overall "*Electrochemical Performance Verification*" section in our **revised manuscript (pages 7–9)**, to incorporate the re-evaluated and additional cell data. Despite these changes, the overall discussion regarding the cyclability and rate capability of the cathodes remains unchanged.

The doping amount of Mg in this work (LCD, SSD, and CPD) is 1.5 mol%, which is not higher than 0.5 wt% (equivalent to 2 mol%), as the reviewer pointed out to be a conventional maximum doping amount. Furthermore, the calcination temperature of cathode materials can vary depending on the synthetic environment and equipment specifications within an appropriate range, which can lead to differences in the optimal temperature. Our synthesis temperature does not deviate significantly from previous reports, and there is considerable number of literatures on the preparation of NCM622 cathode materials at 850 °C (J. Power Sources, 2023, 5553, 232307; J. Electrochem. Soc., 2018, 165, A696; J. Alloys Compd., 2021, 872, 159664; ACS Sustainable Chem. Eng., 2021, 9, 17, 6087-6096) or even lower, as low as 800 °C (Electrochim. Acta, 2015, 182, 795-802; RSC Adv., 2015, 5, 88773-88779). Therefore, we are confident that we have prepared the cathode materials properly.

We appreciate again the valuable comments provided by the reviewer, which have significantly contributed

to enhancing the reliability and quality of our work.

3) NCM622 is more like a “transition” material for the battery industry. For high-Ni cathodes, LiOH·H₂O is required. What is the challenge for making Mg-containing low-grade LiOH·H₂O?

Response: We appreciate the reviewer for bringing up this point. In the original manuscript, we investigate the impact of Mg on Li₂CO₃ production due to its priority as a strategic and universal material. Indeed, the issue of Mg impurities is more pronounced in LH production compared to LC. Impurity elements such as Na⁺ and Mg²⁺ cannot be easily washed out in LH production because LH is also soluble in water. Moreover, a significant loss of LH occurs in the repeated washing process. Thus, the economic and environmental benefit would be greater in LH production when considering these concerns.

[Redacted]

Reply to Reviewer #3

This work discussed the roles of certain impurities such as Mg²⁺ in raw lithium salt and its impact in the as-prepared NMC cathode materials. It tackles the purity and cost issue of raw materials from a different aspect which is of great important to industry. The upper limit of 1%Mg in "not-that-pure" Li₂CO₃ largely relied industry cost in refining brines/rocks when extracting Li₂CO₃. The authors further explored the doping site of Mg provided from three different approaches which is also quite interesting. The techno-economic impacts of this work is well discussed and provides insights for fundament research.

Some questions and suggestions are listed below.

Response: We thank very much for your interest in our work and the kind support for the publication of our work.

1. The authors mentioned that growth of Li₂CO₃ into octahedrons is probably resulted from the cubic morphology of magnetite seeds. Do they have evidence for that? SEM images of magnetite for example?

Response: We are grateful for this valuable suggestion. We attempted to capture SEM images of magnesite; however, it proved challenging to capture the moment of seed generation during LCM synthesis. The

MgCO₃·xH₂O seeds are formed at very early stages of the reaction, resulting in a very small quantity of harvestable product. Additionally, when we extended the reaction time to obtain a larger quantity of the product powder, the MgCO₃·xH₂O particles agglomerated, forming flower-like secondary particles, as shown in the Figure below.

Figure. SEM image of synthesized MgCO₃·xH₂O. The reaction was conducted with 300 ml of solution containing Mg concentration of 1 g L⁻¹ and saturated Na₂CO₃ solution was added with 3 ml min⁻¹ for 30 min.

Instead, we cited the relevant reference [*Phys. Chem. Chem. Phys.*, 2014, 16, 23440-23450] that demonstrates nucleation and growth mechanisms of magnesium carbonate species. Due to the inappropriate specificity of 'magnesite' and 'cubic,' we revised 'magnesite' to 'MgCO₃·xH₂O' and the sentence in our **revised manuscript on page 6–7, line number 24** now reads as follows:

“Li₂CO₃ nucleated and grew on the generated MgCO₃·xH₂O seeds as an octahedron-like morphology, which is highly supposed to have resulted from the MgCO₃·xH₂O seeds²¹.”

Furthermore, the seeding effect of MgCO₃·xH₂O is now more strongly supported by ex-situ PSD analysis (**Supplementary Fig. S7**) and cross-sectional SEM-EDS mapping (**Supplementary Fig. S9**). The LCM maintained a narrower PSD throughout the reaction time and exhibited a single-crystalline morphology without visible pores or grain boundaries.

2. Morphology not only impacts the flowability and compressibility drying powder production, they also change the "filling ratio" of the furnace when synthesizing NMC which is determined by the density of salts. Is there any difference for Li₂CO₃ with different contents of Mg impurities and morphologies?

Response: We thank the reviewer for raising this point. The “filling ratio” is one of the critical factors in the process capacity of cathode material calcination. We roughly tested the tap density of salts by tapping the same weight of LCB, LCM, and LCM-HC in the identical vials. The tap density showed a slight increase due to the presence of Mg impurities, which altered the morphology of Li_2CO_3 , making it more single-crystalline-like (**Supplementary Fig. S11**).

Figure. The filling ratio of cathode material synthesis furnace was tested by roughly comparing the tap density of lithium salts. The 0.3 g of lithium salts were placed into identical 2 ml vials. The tap density exhibited a slight increase with the higher Mg impurity content.

We additionally mentioned about the filling ratio of the furnace in our revised manuscript and marked with highlighter (**page 7, line number 9–11**):

*“Furthermore, the transition to a single-crystalline-like morphology increased the filling ratio of the furnace during the cathode material calcination process, thereby enhancing process capacity (**Supplementary Fig. S11**).”*

3. As the electrochemical performances are conducted in half cells the variation of NMC developed by using different Li_2CO_3 salts are not convincing. Please add some discussion in the paragraph to clarify that in the future full coin cells are still needed to assess the usable capacities and cycling stability. If there are more Mg occupying Li sites, then the usable capacity will be lower in those NMC which, however, cannot be reflected in half cells.

Response: Thanks for this comment. To address this concern, we performed a comprehensive re-evaluation of the electrochemical performance of the cathodes by retesting both half cells and additional full cells at least three times. We acknowledge that the initial capacity test was conducted at a high current density and a low temperature, which could have potentially influenced the initial capacity properties of the cathodes. This suggests that the rate capability properties (**Fig. 3g**) may indeed be intertwined with the initial capacity properties.

We re-evaluated the charge/discharge curves of every cathode material presented in this work at a current rate of 16 mA g⁻¹ with 25 °C. The revised charge/discharge curves are used in **Fig. 3** and **Supplementary Figs. S14 and S19**. In this case, the Bare cathode exhibited the highest initial capacity, possibly due to the redox-inactive nature of Mg²⁺. Furthermore, to support and verify the enhanced performance, we conducted additional high temperature cycling tests and full cell tests (**Fig. 3**). This allowed us to demonstrate the significance of the improvements and further validate the enhanced cyclability observed in the LCD cathode. The consistency in performance (initial capacity and cyclability) was evident not only at 25 °C but also at elevated temperatures of 40 °C and in full cell configurations.

We have revised overall "*Electrochemical Performance Verification*" section in our **revised manuscript (pages 7–9)**, to incorporate the re-evaluated and additional cell data. Despite these changes, the overall discussion regarding the cyclability and rate capability of the cathodes remains unchanged.

4. The roles and impacts of impurities in raw salts was discussed in a recent perspective (Nature Energy, 2023,329-339). Please cite this relevant work.

Response: We thank the reviewer for introducing this recent perspective. We cited the relevant reference (Nat. Energy, 2023, 329-339) in our manuscript with a proper sentence and marked with highlighter (**page 4, line number 8–9**):

“Furthermore, impurity and quality control in industry-scale cathode material production are gaining increased attention¹⁹.”

REVIEWERS' COMMENTS

Reviewer #1 (Remarks to the Author):

The authors have addressed my questions and I am in support of publishing this work.

Reviewer #2 (Remarks to the Author):

I appreciate the authors' efforts and response to my comments. This manuscript may be considered for publication now.

Reviewer #3 (Remarks to the Author):

The authors addressed my questions well. This manuscript is ready for publication.